# Frequent and Durable Anti-HIV Envelope VIV2 IgG Responses Induced by HIV-1 DNA Priming and HIV-MVA Boosting in Healthy Tanzanian Volunteers

**DOI:** 10.3390/vaccines8040681

**Published:** 2020-11-13

**Authors:** Agricola Joachim, Frank Msafiri, Sayali Onkar, Patricia Munseri, Said Aboud, Eligius F. Lyamuya, Muhammad Bakari, Erik Billings, Merlin L. Robb, Britta Wahren, Fred S. Mhalu, Eric Sandström, Mangala Rao, Charlotta Nilsson, Gunnel Biberfeld

**Affiliations:** 1Department of Microbiology and Immunology, Muhimbili University of Health and Allied Sciences, P.O. Box 65001 Dar es Salaam, Tanzania; frankmbulinyingi@yahoo.com (F.M.); aboudsaid@yahoo.com (S.A.); eligius_lyamuya@yahoo.com (E.F.L.); mhalufred@yahoo.com (F.S.M.); 2Division of Clinical Microbiology, Department of Laboratory Medicine, Karolinska Institutet, 17177 Stockholm, Sweden; charlotta.nilsson@ki.se; 3The US Military HIV Research Program, The Henry M Jackson Foundation for the Advancement of Military Medicine, Bethesda, MD 20817, USA; sayali.onkar@pitt.edu (S.O.); ebillings@childrensnational.org (E.B.); 4United States Military HIV Research Program, Walter Reed Army Institute of Research, Silver Spring, MD 20910, USA; mrobb@hivresearch.org (M.L.R.); mrao@hivresearch.org (M.R.); 5Department of Internal Medicine, Muhimbili University of Health and Allied Sciences, P.O. Box 65001 Dar es Salaam, Tanzania; pmunseri@yahoo.com (P.M.); drbakari@yahoo.com (M.B.); 6Henry M Jackson Foundation for the Advancement of Military Medicine, Bethesda, MD 20817, USA; 7Department of Microbiology, Tumor and Cell Biology, Karolinska Institutet, 17177 Stockholm, Sweden; britta.wahren@ki.se; 8Venhälsan, Karolinska Institutet at Södersjukhuset, 11883 Stockholm, Sweden; eric.g.sandstrom@gmail.com; 9Department of Microbiology, Public Health Agency of Sweden, 17182 Solna, Sweden; 10Department of Global Public Health, Karolinska Institutet, 17177 Stockholm, Sweden; gunnel.biberfeld@ki.se

**Keywords:** HIV, vaccine, DNA, MVA, V1V2 antibodies

## Abstract

We evaluated antibody responses to the human immunodeficiency virus (HIV) envelope variable regions 1 and 2 (V1V2) in 29 vaccinees who had received three HIV-1 DNA immunizations and two HIV-modified vaccinia virus Ankara (MVA) boosts in the phase I/II HIVIS03 vaccine trial. Twenty vaccinees received a third HIV-MVA boost after three years in the HIVIS06 trial. IgG and IgG antibody subclasses to gp70V1V2 proteins of HIV-1 A244, CN54, Consensus C, and Case A2 were analysed using an enzyme-linked immunosorbent assay (ELISA). Cyclic V2 peptides of A244, Consensus C, and MN were used in a surface plasmon resonance (SPR) assay. Four weeks after the second HIV-MVA, anti-V1V2 IgG antibodies to A244 were detected in 97% of HIVIS03 vaccinees, in 75% three years later, and in 95% after the third HIV-MVA. Anti-CN54 V1V2 IgG was detectable in 48% four weeks after the second HIV-MVA. The SPR data supported the findings. The IgG response was predominantly IgG1. Four weeks after the second HIV-MVA, 85% of vaccinees had IgG1 antibodies to V1V2 A244, which persisted in 25% for three-years. IgG3 and IgG4 antibodies to V1V2 A244 were rare. In conclusion, the HIV-DNA/MVA vaccine regimen induced durable V1V2 IgG antibody responses in a high proportion of vaccinees.

## 1. Introduction

Despite tremendous success in reducing acquired immunodeficiency syndrome (AIDS) related deaths and new HIV infections, existing HIV prevention strategies have not succeeded in stopping transmission of HIV [1]. The availability of an efficacious HIV vaccine would contribute significantly in halting transmission and eliminating HIV/AIDS [2,3]. So far, only seven HIV-1 vaccine efficacy trials have been conducted [4,5,6,7,8,9,10,11]. The RV144 HIV vaccine efficacy trial showed a 31.2% vaccine efficacy against HIV infection in healthy individuals primed with ALVAC-HIV vCP1521 and boosted with AIDSVAX B/E at 42 months after vaccination [9]. Post hoc analysis estimated a cumulative vaccine efficacy of 60.5% at 12 months that decreased over time with the decay in immune responses [12]. The early analysis of the immune correlates of infection risk in the RV144 vaccinees demonstrated that IgG antibodies binding to the variable regions 1 and 2 (V1V2) of the HIV envelope (Env) correlated inversely with the risk of HIV-1 acquisition, while high levels of plasma IgA antibodies against Env correlated with an increased risk of HIV infection [13]. Sieve analysis suggested that V1V2 antibodies are capable of selectively blocking specific HIV-1 variants, further supporting vaccine-induced V2 responses as having a preventive role [14]. Additionally, V1V2-specific monoclonal antibodies have been correlated with control or protection from HIV, SIV and SHIV [15,16].

HIV-1 preferentially targets activated CD4^+^ T cells that, in addition to CD4 and CCR5 receptors, express α4β7 receptors [17]. The conserved motif within the V2 loop mediates the binding of HIV-1 Env to α4β7, leading to virion capture on the surface of CD4^+^ T cells [18]. Although α4β7 integrin is not required for entry of the virus into CD4^+^T cells [18], it is believed that induction of anti-V2 responses that inhibit the formation of α4β7-gp120 complex will significantly reduce the risk of HIV acquisition [18,19]. Moreover, IgG binding antibodies to V1V2 of HIV-1 Env have been reported to inhibit viral entry, possibly by blocking the binding of V1V2 to the integrin α4β7 receptor [20,21].

The IgG subclasses differ in their affinities for Fc receptors of which IgG1 and IgG3 interact more efficiently with most Fc receptors than IgG2 and IgG4 [22]. Hence, different IgG subclasses mediate different effector functions [22,23]. IgG3 antibodies are more efficient at neutralizing cell free virus and blocking HIV-1 Env-mediated cell fusion than IgG1 and IgG2 [24]. High IgG1 and IgG3 antibody responses to HIV-1 V1V2 were induced in the RV144 trial [25], and V1V2 specific IgG3 responses correlated with reduced risk of HIV infection among vaccinees [26]. Pathogen-specific IgG3 has also been reported to correlate with protection in other infectious diseases, like malaria and Chikungunya virus infection [27,28].

We have previously reported frequent humoral and cellular immune responses in vaccinees given three HIV-DNA and two HIV-modified vaccinia virus Ankara (MVA) immunizations in the HIVIS03 trial [29,30]. Twenty HIVIS03 vaccinees were given a third HIV-MVA immunization three years after the second HIV-MVA immunization (the HIVIS06 trial) [31]. Four weeks after the second HIV-MVA, all of 20 vaccinees had binding antibodies to subtype C gp140 and 89% had antibodies to subtype B gp160. None had demonstrable neutralizing antibodies in a peripheral blood mononuclear cell assay using subtype B SF162 or CRF01_AE CM244 virus nor in a TZM-bl based assay using subtype B SF162 or subtype C MW965.26 pseudovirus [29]. The majority, 95% and 88%, had detectable antibody dependent cell-mediated cytotoxicity (ADCC) activity to CRF01_AE using infected cells and CRF01_AE gp120 coated cells, respectively. IFN-γ ELISpot responses to Gag and Env were detected in 89% of the vaccinees. Three years after the second HIV-MVA, binding antibody responses to subtype C gp140 and subtype B gp160 were detected in 90% and 85% respectively and the majority (84%) had ADCC–mediating antibodies to CRF01_AE. IFN-γ ELISpot responses were demonstrated in 74% of the vaccinees. An increase in Env-binding antibodies and ADCC-mediating antibodies was demonstrated after a third HIV-MVA immunization [31].

In the present study, given the fact that the HIV-DNA/MVA vaccine recipients in the HIVIS03/06 trials developed potent and durable immune responses [30,31], we explored anti-Env V1V2 antibody responses.

## 2. Materials and Methods

### 2.1. Ethics Statement

The HIVIS03 and HIVIS06 trial protocols were approved by Tanzania’s National Health Research Ethics Committee and the Senate Research and Publications Committee of the Muhimbili University of Health and Allied Sciences (MUHAS), and the Tanzania Food and Drugs Authority. Approvals were received from the Regional Ethics Committee, Stockholm, Sweden. Both trials were conducted in accordance with the International Conference on Harmonization and Good Clinical Practice guidelines. All volunteers provided a written informed consent before enrollment into the study. The in vitro assays that were carried out at the Walter Reed Army Institute of Research (WRAIR) were conducted under the Institutional Review Board approval, WRAIR # 2219/RV444.

### 2.2. Study Design

In the HIVIS03 trial, 60 HIV-uninfected volunteers had been randomized into three groups with 20 volunteers each to receive placebo or 1 mg HIV-DNA intradermally or 3.8 mg intramuscularly. HIV-DNA plasmids expressing HIV-1 gp160 subtypes A, B, C; Rev B; Gag A, B and RTmut B were given at months 0, 1, and 3 using a needle-free Biojector device. Recombinant MVA expressing CRF01_AE HIV-1 Env and Gag-Pol subtype A (MVA-CMDR [32], here referred to as HIV-MVA), was administered intramuscularly by needle at months 9 and 21 (Figure 1) [29]. At the completion of the HIVIS03 trial, participants were asked to participate in a follow-up trial (HIVIS06). In the HIVIS06 trial, 20 volunteers, who had received all five immunizations in the HIVIS03 trial, were given a third HIV-MVA, three years after the second HIV-MVA (Figure 1) [31]. In the present study, stored plasma samples from HIVIS03 and HIVIS06 vaccinees, and placebo recipients were used. The samples had been collected at baseline before immunization and four weeks after the second HIV-MVA (*n* = 29 vaccinees and 12 placebo recipients), at the time of the third HIV-MVA, i.e., three years after the second HIV-MVA boost and four weeks after the third HIV-MVA vaccination (*n* = 20 vaccinees).

### 2.3. Assessment of IgG Binding Antibodies to Scaffolded gp70 V1V2 Region

Testing of IgG binding antibodies to scaffolded gp70 V1V2 protein was performed using enzyme-linked immunosorbent assay (ELISA). Immulon 2U-bottom microtiter plates (Thermo Scientific, Rochester, New York, NY, USA) were coated with 100 μL (0.2 μg/well) of recombinant gp70 (MLV)-V1V2 protein (CRF01_AE A244, subtype C CN54, subtype C Consensus C, or subtype B Case A2, Immune Technology Corp., New York, NY, USA) diluted in PBS overnight at 4 °C and blocked with 200 μL/well of blocking buffer. The plates were incubated for 1 h at room temperature with 100 μL of serial dilutions of plasma starting at 1:100 dilution (in triplicate). Sheep anti-human IgG-HRP (Binding Site, Birmingham, UK) was used at a 1:1000 dilution. ABTS Peroxidase Substrate A and Peroxidase Solution B (KPL, Gaithersburg, MD, USA) were used for visualization. Absorbance was read at 405 nm after 1 h incubation in the dark. Each plate included a HIV positive and a negative control. Results are expressed as end point titers, defined as the highest reciprocal dilution that yielded an absorbance twice the baseline values.

### 2.4. Assessment of V1V2-Specific IgG Subclasses

IgG1, IgG2, IgG3 and IgG4 antibodies to scaffolded gp70 (MLV)-V1V2 proteins were determined using ELISA as described above for total IgG using sheep anti-human IgG1-HRP, IgG2-HRP, IgG3-HRP, and IgG4-HRP (Binding Site, Birmingham, UK).

### 2.5. Surface Plasmon Resonance (SPR) Assay

Measurements were performed using a Biacore T100 as previously described [33]. Briefly, lysozyme and streptavidin were immobilized onto CM5 chips. Cyclic V2 biotinylated peptides representing A244 CRF01_AE, subtype C Concensus C or subtype B MN (1 μM) were manually injected over the streptavidin-coated chip surface. Plasma samples were heat inactivated at 56 °C for 45 min. Diluted plasma samples (1:50) were injected over the chip surface followed by a dissociation period. Thereafter, a 50 nM solution of affinity-purified gamma chain-specific sheep anti-human IgG was passed over the peptide coated-Ig bound surface. Non-specific binding was subtracted and data analysis was performed using BIA evaluation 4.1 software. The reported response units (RU) for the IgG specific values are the difference between the average value of a 5 s window taken 60 s after the end of the anti-IgG injection and the one taken at 10 s before the beginning of the anti-IgG injection. All samples were run in triplicate.

### 2.6. Data Analysis

Data were analyzed using GraphPad PRISM version 7. Pairwise analysis was employed to compare anti-V1V2 responses between different immunization time points. A McNemar test was used for comparison of frequency of V1V2 responses between different immunization time points, and Wilcoxon matched-paired signed rank test for comparison of magnitude of antibody responses. A two-tailed *p*-value < 0.05 was statistically significant.

## 3. Results

### 3.1. IgG Binding Antibodies to the VIV2 Region of HIV-1 Envelope

Anti-Env-VIV2 antibodies were measured in plasma samples collected from 29 vaccinees and 12 placebo recipients at baseline (HIVIS03 visit 3, V3) and four weeks after the second HIV-MVA (HIVIS03 visit 21, V21). Four weeks after the second HIV-MVA, 28/29 (97%) vaccinees had IgG binding antibodies to V1V2 of CRF01_AE A244, 14/29 (48%) to subtype C CN54, and 3/29 (10%) to subtype B Case A2. None of the vaccinees had IgG binding antibodies to V1V2 of subtype C consensus C (Table 1).

No anti-V1V2 reactivity was detected among placebo recipients to any of the subtypes tested (data not shown).

V1V2-specific antibody responses were also determined in 20 vaccinees who received a third HIV-MVA. The data is summarized in Table 2. The plasma sample from one volunteer was not available in sufficient volume for VIV2 IgG binding testing four weeks after the second HIV-MVA boost. All 19 (100%) vaccinees had IgG binding antibodies to V1V2 of CRF01_AE A244 with median titers of 3200 (IQR; 1600–12800) four weeks after the second HIV-MVA (Table 2 and Figure 2A). At the time of the third HIV-MVA boost, the majority of the vaccinees, 15 out of 20 (75%) still had anti-V1V2 CRF01_AE A244 responses with median titer of 300 (IQR 50–800) (Table 2 and Figure 2A). Four weeks after the third HIV-MVA, the response rate to V1V2 of CRF01_AE A244 increased to 19 in 20 (95%), *p* = 0.125 and the magnitude was significantly enhanced by the third HIV-MVA boost, with median titers rising to 1600 (IQR; 800–3200) *p* < 0.0001 (Table 2 and Figure 2A). Of the 20 vaccinees, nine had received HIV-DNA vaccine intradermally and 11 intramuscularly. The frequency and magnitude of the V1V2 total IgG antibody responses were not significantly different between these two groups at any of the time points tested (data not shown).

In contrast to the V1V2 CRF01_AE A244-specific responses, the response rate to V1V2 of subtype C CN54 decreased significantly three years after the second HIV-MVA. At the time of the third HIV-MVA, the proportion of vaccinees with antiV1V2 antibodies to subtype C CN54 had declined from 47% (9 out of 19) to 10% (two in 20), *p* = 0.016. The magnitude in the nine responders was 800 (IQR; 400–2400) four weeks after the second HIV-MVA. The third HIV-MVA did not significantly boost neither the frequency nor the level of subtype C CN54 V1V2-specific IgG antibodies, *p* = 0.5 (Figure 2B). IgG binding antibody responses to V1V2 of subtype B Case A2 were rare (Figure 2C), and the vaccine did not elicit detectable IgG binding antibodies to subtype C Consensus C V1V2 (Table 2). 

### 3.2. Vaccine-Induced Anti-VIV2 IgG1, IgG2, IgG3, and IgG4 Responses

Anti-A244 V1V2 responses were primarily IgG1 (Table 2). Four weeks after the second HIV-MVA boost, 17 out of 20 (85%) of vaccinees had anti-A244 V1V2 IgG1 antibodies. The V1V2 IgG1 responses declined significantly over time, but were still detectable in five out of 20 (25%) of vaccinees three years later. The response rate of IgG1 binding antibodies to A244 V1V2 was significantly boosted by the third HIV-MVA to 13 in 20 (65%), *p* = 0.008 (Figure 3A). IgG2 binding antibodies to A244 V1V2 were not detectable. The proportion of vaccinees with IgG3 binding antibodies to A244 V1V2 was low, detectable in three out of 20 (15%) four weeks after the second HIV-MVA. Anti-A244 V1V2 IgG3 responses were undetectable three years later (Figure 3B), and was only detectable in one of 20 vaccinees after the third HIV-MVA. IgG4 binding antibodies to A244 V1V2 were only demonstrable in one out of 20 (5%) four weeks after the second HIV-MVA boost.

The overall magnitude of anti-CRF01_AE A244 V1V2 IgG1 responses declined significantly from a median of 400 (IQR; 125–800) at peak immunogenicity after the second HIV-MVA to nearly undetectable levels three years later (*p* < 0.0001). In the five responders, the median titer was 100 (IQR: 100–800), three years after the second HIV-MVA. Anti-A244 V1V2 IgG1 titers were not significantly increased by the third HIV-MVA immunization among responders, a median titer 200 (IQR; 100–500) four weeks after the third HIV-MVA (*p* = 0.251) (Figure 3A).

IgG1 subclass determination of antibodies to subtype C CN54 V1V2 and subtype B Case A2 V1V2 was performed only on samples exhibiting V1V2 total IgG reactivity. Anti-subtype C CN54 V1V2 IgG1 responses were detectable in three in nine (33%) of vaccines four weeks after the second HIV-MVA, undetectable three years later and detectable in two in nine (22%) after the third HIV-MVA (Table 2). IgG1 binding antibodies to Case A2 V1V2 were not detected in the three vaccinees tested (Table 2).

### 3.3. Antibody Response to Cyclic V2 Peptide

The SPR analysis demonstrated IgG antibodies specific to both subtype E A244 and consensus C cyclic V2 peptides four weeks after the second HIV-MVA. At the three-year time point, only IgG responses to CycV2 A244 were still detectable (Figure 4). The response rate was significantly boosted by the third HIV-MVA immunization (*p* < 0.0001). No antibody reactivity was observed using HIV MN subtype B cyclic V2 peptides. There was no reactivity detected among placebo recipients to any of the cyclic V2 peptides tested (Figure 4).

### 3.4. Correlation between V1V2 Antibodies and ADCC-Mediating Antibody Responses

We also determined the association between V1V2 antibodies and previously reported ADCC-mediating antibodies from the same vaccinees [31]. There was no correlation between the magnitudes of V1V2 IgG antibodies or V1V2 IgG1 antibodies to AE A244 and ADCC-mediating antibodies against CRF01_AE gp120 coated cells and CRF01_AE virus-infected cells four weeks after the second HIV-MVA boost (Table 3).

## 4. Discussion

In the current study, we investigated the capability of an HIV-DNA prime HIV-MVA boost vaccine to elicit antibody responses to the VIV2 region of HIV-1 Env, and the three-year durability of the induced anti-V1V2 antibody responses. At peak immunogenicity, the HIV-DNA/MVA vaccine regimen elicited IgG binding antibodies to V1V2 of CRF01_AE A244 in 97% of the vaccinees and subtype C CN54 in 48%, while antibodies to V1V2 of subtype B were rare. Durable IgG binding antibodies against V1V2 of CRF01_AE A244 were induced in the majority of the vaccinees, persisting for three years in 75% of the vaccine recipients. The anti-V1V2 antibody titers were quite low at this time point. The third HIV-MVA immunization at the three-year time point increased the frequency of anti-CRF01_AE A244 V1V2 IgG antibody responses to 95%.

This is the first study demonstrating a three-year durability of V1V2 antibody responses in HIV-DNA primed and HIV-MVA boosted vaccinees. In the RV144 trial, longitudinal follow up of V1V2-specific IgG responses between peak immunogenicity at week 26 to 182 revealed a rapid decline of V1V2 IgG response rates. At the time of peak immunogenicity (week 26), 97% of the vaccinees had IgG antibodies to V1V2 of subtype C (C.1086) and subtype B (Case A2) [26]. One year after peak immunogenicity (week 78), IgG response rates fell to 24% and 3% for C.1086 gp70V1V2 and gp70V1V2 CaseA2, respectively [26]. Subsequently, 162 HIV uninfected RV144 vaccinees were enrolled in the RV305 follow on trial and received two additional boosts of AIDSVAX B/E with or without ALVAC-HIV, 6–8 years after the last RV144 vaccination [34]. Weak residual antibody responses were detected before administration of additional boosts [34]. The late boosts significantly enhanced IgG binding antibody responses against gp70 V1V2 92TH023 and gp70 V1V2 Case A2 compared to two weeks after the last RV144 vaccination. However, the boosted immune responses were sustained for less than six months [34]. Six-month durability of V1V2 binding IgG antibodies has recently been reported in HVTN 105 vaccinees. The HVTN 105 vaccine trial was a randomized, double blind, clinical trial that assessed the safety and immunogenicity of vaccine regimens using a DNA-HIV vaccine of subtype C given together with AIDSVAX B/E [35]. In vaccinees who had received both DNA and AIDSVAX B/E four times during six months, antibody responses were demonstrated in 100% of vaccinees to A244AE V1V2 and 71% to 1986.C V1V2 six months after the last vaccination [35].

In the present study, using scaffolded gp70 V1V2 protein in ELISA, IgG1 dominated over other IgG subclasses. CRF01_AE A244 V1V2 IgG1 antibodies were detectable in 85% of vaccinees four weeks after the second HIV-MVA vaccination, and the responses lasted for three years in 25% of them. In contrast, anti-A244 V1V2 IgG3 antibodies were only detected in 15% of the vaccinees and undetectable three years later. A similar predominance of vaccine-induced V1V2 IgG1 over V1V2 IgG3 responses was also observed in the RV144 and VAX003 HIV vaccine efficacy trials [26,36]. In the VAX003 trial, rgp120 HIV-1 Env immunogens from subtypes B MN and CRF01_AE A244 strains were assessed for vaccine efficacy against HIV-1 infection [5] and Env proteins alone did not prevent HIV-1 infection nor delay the onset of AIDS in infected VAX003 vaccinees [5]. The frequency of IgG3 binding antibodies to V1V2 was higher in the RV144 participants compared to the VAX003 vaccine recipients [25]. Although IgG3 response rates to V1V2 correlated with decreased risk of HIV infection in the RV144 trial, their durability was short lived. Weak IgG3 binding antibody responses to clade C C.1086 V1V2 tags, and gp70 B.CaseA2 V1V2 were detected 6.5 months (week 52) after peak immunogenicity [26]. It is worth noting that, in normal plasma, IgG1 contributes to 60% of total IgG, whereas IgG3 contributes to 10% of total IgG [37]. Furthermore, the half-life of IgG3 is reported to be substantially shorter (nine days) than that of the other subclasses (23 days) [38], which may explain the differences in durability between IgG1 and IgG3 specific for V1V2 following vaccination. In our study, total IgG V1V2 responses did not predict IgG subclass responses. Similarly, Yates et al. have reported that vaccine-induced IgG and IgG3 V1V2 antibody responses were not strongly predictive of one another [26].

Here, the HIV-DNA/MVA regimen elicited lower IgG binding response rates to scaffolded gp70 VIV2 CN54 (subtype C) than to the V1V2 loop of CRF01_AE A244. Forty-seven percent of vaccinees had IgG antibody responses to subtype C CN54 V1V2 four weeks after the second HIV-MVA. The responses were sustained for three years in only 10% of the vaccinees, at a low magnitude. Few vaccinees (16%) had anti-V1V2 Case A2 (subtype B) antibodies at peak immunogenicity. Since the MVA-CMDR insert is based on an AE isolate sequence the superior frequency and magnitude of responses to AE recombinant proteins and peptides is expected. Previous studies have shown that HIV vaccine regimens can induce varying response rates to different V1V2 immunogens. The RV144 regimen elicited higher IgG1 responses to 92TH023 V1V2 subtype E Env scaffold than to subtype B Case A2 V1V2, while in the non-protective VAX004 HIV efficacy trial [4], the vaccine regimen of subtype B/E immunogens stimulated higher IgG1 antibody responses to subtype B Case A2 V1V2 than to CRF01_AE V1V2 [36]. Additionally, 100% of recipients of the PENNVAX-G DNA (containing subtypes A, C, and D)/MVA-CMDR prime-boost regimen developed IgG antibody responses to gp70V1V2 CRF01_AE 92TH023 whereas antibodies to gp70V1V2 subtype B Case A2 were rare [39]. Notably, at peak immunogenicity, none of the African recipients (0/77) of PENNVAX-G DNA/MVA-CMDR vaccine regimen generated antibodies to subtype B gp70V1V2 Case A2 [39]. In the HVTN 094 phase 1 trial recipients of two immunizations of the GeoVax subtype B DNA vaccine followed by two or three boosts with a MVA vaccine of subtype B more than 90% of vaccinees developed binding IgG Env antibodies to gp140 and gp41 at peak immunogenicity and with a durability of 12 months post vaccination. IgG reactivity to gp70V1V2 Case A subtype B was demonstrated in 20% of vaccinees who received two MVA boosts and in 21% of vaccinees who received three MVA boosts [40]. In a study applying an Indian subtype C DNA prime (ADVAX) and a recombinant MVA (TBC-M4) boost in the P001 vaccine trial, antibodies to V2 peptides were detected in (50%) of the vaccinees after the delivery of two ADVAX at zero and one month and TBC-M4 at three and six months [41].

IgG binding to cyclicV2 (CycV2) peptides were detected in the majority of the vaccinees in the present study. The antibody responses were predominantly directed against CycV2 A244, less so to CycV2 Consensus C, and none to CycV2 subtype B MN, which resembled the response pattern detected by using scaffolded proteins in the ELISA. In the RV144 trial, IgG responses to CycV2 92TH023 were demonstrated, but not to subtype B Case A2 [36].

V2 antibodies isolated from RV144 vaccinees have been shown to synergize with C1 antibodies for infectious virus capture, neutralization and ADCC [42]. In the present study, we did not find a correlation between the magnitude of ADCC-mediating antibodies and V1V2 specific IgG antibodies, indicating that these antibody responses may target different epitopes on the HIV-1 Env. In the RV144 vaccine trial, ADCC- activity was correlated with reduced risk of infection in vaccinees with low levels of Env-specific IgA antibodies [13]. V2 monoclonal antibodies (CH58 and CH59) from RV144 vaccinees have been shown to mediate ADCC [43]. Following late boosting of RV144 vaccinees in the RV305 trial, additional vaccine-induced V2-specific monoclonal antibodies were characterized showing epitope specificities different from monoclonal antibodies CH58 and CH59 with increased antibody-mediated effector functions, including ADCC [44].

Although the V1V2 antibody responses detected in the present study were long-lasting, the breadth was limited to two of three of the V1V2 Env subtypes tested. IgG binding antibody responses were elicited against V1V2 of CRF01_AE A244 in 97% and against subtype C CN54 antigen in 48% of the vaccinees but were rare (10%) against subtype B case A2 antigen. We have previously reported a high frequency of ELISA antibody reactivity to subtype B IIIB gp160 (90%), HIV-1_96ZM651_subtype C gp140 (100%) and CRF01_AE CM234 gp120 (100%) in the same vaccinees four weeks after the second HIV-MVA [30]. The HIV-DNA priming vaccine contained plasmids expressing HIV-1 gp160 subtypes A, B, C, Rev B, Gag A, B, and RTmut B, and the recombinant HIV-MVA boost expressed CRF01_AE HIV-1 Env subtype E and Gag-Pol subtype A [29]. The demonstration of IgG V1V2 responses to subtype C CN54 in 48% of vaccinees suggests that the HIV-DNA immunizations may have primed for the induction of subtype C antibody responses. The observed low frequency of subtype B V1V2 antibody responses is intriguing, but may be attributed to the absence of Env immunogens inducing antibodies recognizing Case A2 V1V2 or MN.

The study was limited to the number of vaccinees who completed all vaccinations in the HIVIS03 trial. Of the 40 vaccine recipients, only 30 completed the immunization schedule but plasma/serum samples for analysis were only available from 29 vaccinees [29,31]. Of these, 20 participated in the HIVIS06 trial and received a third HIV-MVA boost. We recently reported V1V2 binding antibody responses to CRF01_AE A244 in 25 (81%) of 31 healthy African vaccinees at peak immunogenicity after receiving three HIV-DNA vaccinations and two HIV-MVA boosts in the TaMoVac II trial [45]. Further study of the durability of vaccine-induced V1V2 antibodies is warranted.

The levels of Env-specific IgA antibodies in plasma, which were shown to correlate with increased risk of HIV infection in the RV144 trial [13], were not assessed in this study. However, in another HIV vaccine trial using three immunizations with the same HIV-DNA and two immunizations with the same HIV-MVA as used in the present study, plasma IgA antibodies to Env were elicited in only 3% of the vaccinees [45].

In this study, the late 3rd HIV-MVA increased the anti-V1V2 A244 responses three years post the second HIV-MVA although not to the same rate as post the second HIV-MVA. This indicates that the anti-V1V2 responses can be maintained by regular boosting with HIV-MVA even in the presence of vector-specific antibody responses as demonstrated earlier for binding antibodies to gp140 and gp160, and ADCC-mediating antibodies in the same vaccinees [31].

A Phase IIb HIV prophylactic double-blind vaccine trial (PrEPVacc) involving high risk HIV uninfected individuals from four countries in East and Southern African is projected to begin late 2020 [46]. The PrEPVacc trial will evaluate the effectiveness of combining pre-exposure prophylaxis with HIV vaccines in reducing HIV acquisition. Two experimental combination vaccine regimens with placebo control will be compared in a three-arm, two-step randomization trial. Participants will be primed with DNA/AIDSVAX at weeks 0, 4, 24, and 48 or DNA/CN54gp140 (weeks 0 and 4) and boosted with MVA/CN54gp140 vaccines at weeks 24 and 48. Furthermore, the same participants will be randomized to receive either TDF/FTC (Truvada) or TAF/FTC (Descovy) daily as pre-exposure prophylaxis [46].

## 5. Conclusions

The HIV-DNA prime HIV-MVA boosting vaccination strategy induced durable IgG antibody responses to VIV2 of CRF01_AE A244 in a high proportion of vaccinees. Furthermore, at peak immunogenicity, anti-V1V2 subtype C responses were also detectable. The findings support the further development and evaluation of DNA prime/MVA boost HIV vaccine regimens.

## Figures and Tables

**Figure 1 vaccines-08-00681-f001:**
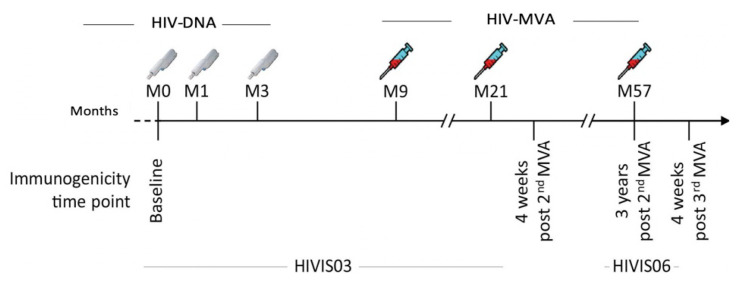
Vaccination schedule and immunogenicity follow up time points for HIVIS03 and HIVIS06 trials. HIV-DNA priming immunizations were given using a needle-free injection device, 1 mg intradermally or 3.8 mg intramuscularly. HIV-MVA boosting immunizations were delivered intramuscularly using needle and syringe. Each boosting vaccination contained 1 mL of 10^8^ plaque-forming units (pfu) of recombinant HIV-MVA vaccine. Testing was done at baseline, four weeks after the second HIV-MVA boost, at the time of the third HIV-MVA boost and four weeks after the third HIV-MVA boost.

**Figure 2 vaccines-08-00681-f002:**
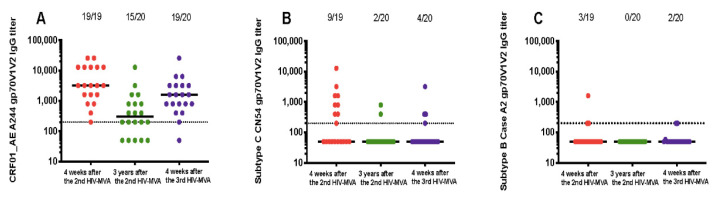
Durability of binding IgG responses to V1V2 scaffolds. Response rates against gp70V1V2 region of CRF01_AE A244 (**A**) subtype C CN54 (**B**) and subtype B Case A2 (**C**) as detected by ELISA at four weeks post second HIV-MVA vaccination (V21, response in red circles), three years after the second HIV-MVA immunization (V62, response in green circles) and four weeks after the third HIV-MVA boost (V64, response in purple circles). Horizontal solid lines represent the median titers of antibody responses at different time points, while dotted line indicates a cut off for positive values. For graphing purposes, negative samples were arbitrarily given a value of 50.

**Figure 3 vaccines-08-00681-f003:**
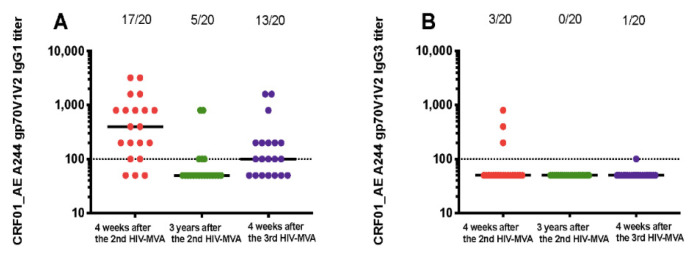
IgG subclass responses to gp70V1V2 protein of CRF01_AE A244. Vaccine-induced IgG1 (**A**) and IgG3 (**B**) binding responses to CRF01_AE A244 V1V2 antigen. Antibody responses at four weeks post second HIV-MVA vaccination (V21, response in red circles), three years after the second HIV-MVA immunization (V62, response in green circles) and four weeks after the third HIV-MVA boost (V64, response in purple circles). Horizontal solid lines represent the median antibody titers, while dotted line indicates a cut off for positive values. For graphing purposes, samples with no V1V2 IgG1 and IgG3 binding antibody responses at 1:100 dilutions were arbitrarily assigned a value of 50.

**Figure 4 vaccines-08-00681-f004:**
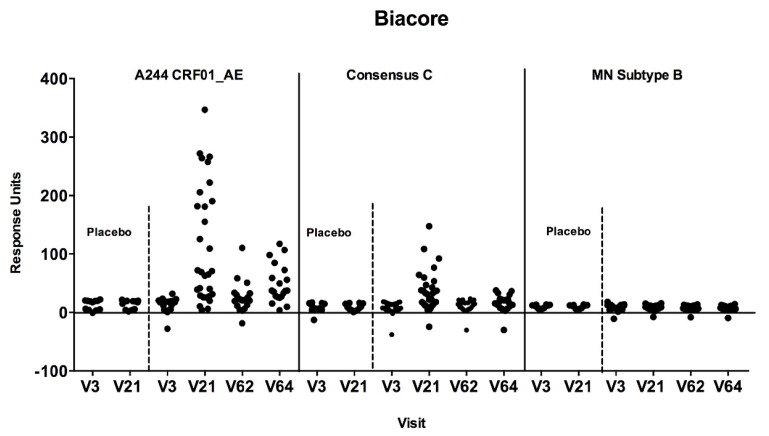
Antibody responses to cyclic V2 loop peptides by SPR/Biacore assay. Plasma samples were tested against A244 CRF01_AE, Consensus C and MN subtype B at baseline (V3) four weeks after the second HIV-MVA (V21), at the time of the third (V62), and four weeks after the third HIV-MVA boost (V64). Plasma samples were used at a 1:50 dilution and the values are reported as response units. Responses were considered positive if they significantly (*p* < 0.05) exceeded the response units at baseline.

**Table 1 vaccines-08-00681-t001:** Frequency of V1V2 IgG antibodies four weeks after the second HIV-MVA vaccination (HIVIS03 trial).

Antibody	Antigen gp70V1V2	Subtypes	Positive/Total Number Tested (%)
**IgG**	A244	AE	28/29 (97)
	CN54	C	14/29 (48)
	Case A2	B	3/29 (10)
	Consensus C	C	0/29

**Table 2 vaccines-08-00681-t002:** Frequency of V1V2 IgG antibodies and IgG subtypes in 20 recipients of a late third HIV-MVA boost given three years after the second HIV-MVA vaccination.

Ab	Antigen gp70V1V2	Subtypes	Positive/Total Number Tested (%)				
			Four Weeks after the 2nd HIV-MVA ^a^	At the Time of the 3rd HIV-MVA ^b^	Four Weeks after the 3rd HIV-MVA ^c^	a vs. b	b vs. c
Total IgG	A244	AE	19/19 (100)	15/20 (75)	19/20 (95)	0.125	0.125
	CN54	C	9/19(47)	2/20 (10)	4/20 (20)	0.016	0.5
	Case A2	B	3/19 (16)	0/20	2/20 (10)	0.25	0.5
	Consensus C	C	0/20	0/20	0/20		
IgG1	A244	AE	17/20 (85)	5/20 (25)	13/20 (65)	0.000	0.008
IgG2	A244	AE	0/20	0/20	0/20		
IgG3	A244	AE	3/20 (15)	0/20	1/20 (5)	0.25	1
IgG4	A244	AE	1/20 (5)	0/20	0/20		
IgG1	CN54	C	3/9 (33)	0/9	2/9 (22)	0.25	0.5
	Case A2	B	0/3	0/3	0/3		

(^a^) Response rates of vaccinees who generated V1V2-specific antibodies four weeks after the second HIV-MVA vaccination, (^b^) at the time of the third HIV-MVA, three years after the second HIV-MVA vaccination and (^c^) four weeks after the third HIV-MVA vaccination. McNemar test was used for comparison of frequencies. *p*-values below <0.05 were considered significant. Ab: Antibody.

**Table 3 vaccines-08-00681-t003:** No correlation between the magnitude of V1V2 antibodies and ADCC-mediating antibodies four weeks after the second HIV-MVA vaccination.

Antibody	Antigen	ADCC	ADCC
	**gp 70V1V2**	**gp120 coated cells** (CRF01_AE CM235)	**IMC-infected cells** (CRF01_AE CM 243)
		**r * (*p*-value)**	**r * (*p*-value)**
IgG	A244 CRF01_ AE	0.113 (0.645)	0.124 (0.614)
IgG1	A244 CRF01_ AE	0.343 (0.156)	0.236 (0.331)

* Spearman coefficient correlation; ADCC, antibody-dependent cellular cytotoxicity; IMC, infectious molecular clone.

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
