# Peer review of "Frequent and Durable Anti-HIV Envelope VIV2 IgG Responses Induced by HIV-1 DNA Priming and HIV-MVA Boosting in Healthy Tanzanian Volunteers"

_vaccines, 2020, doi:10.3390/vaccines8040681_

Round 1

Reviewer 1 Report

Joachim et al. describe in this study the influence of a late MVA boost on V1V2-specific antibody responses in a DNA/MVA prime/boost HIV vaccine trial.

Specific comments:

  • Line 168-175 + Table 1: Authors describe that 29 vacinees and 12 placebo recipients were analyzed for V1V2-specific antibodies. However, only results for the vaccines are shown. One can assume that all controls were negative. However, this should be mentioned in the text or table.
  • In Table 2 total IgG response and responses for subtypes against A244 V1V2 are shown. While total IgG was well maintained after 3 years, responses to all 4 subtypes strongly declined. Authors should comment on this discrepancy.
  • Table 2: “Positive/Total Number Tested (5)” should be changed to “Positive/Total Number Tested (%)”.
  • Authors should consider exchanging V21, V62 and V64 labeling in Figure 2, 3 and 4 with 4 weeks post second HIV-MVA, 3 years post second HIV-MVA and 4 weeks post third HIV-MVA. This would make figures much easier to understand without the need of reading the captions.
  • Line 227/228: Authors state that anti-A244 V1V2 IgG1 titers were not enhanced by the third HIV-MVA immunization. The authors should clarify this statement as responses 4 weeks after the third MVA immunization only were not increased compared to 4 weeks after the second MVA immunization. However, they were significantly increased compared to the 3 years after the second MVA time point.
  • The authors conclude from their study that the DNA /MVA prime/boost regiment should be further explored. However, they could not detect any neutralizing antibodies in an earlier analysis of this study. Here they show that they have a lasting V1V2-specific response which could mediate protection. However, the breath of the V1V2-specific response in their study is rather limited. They should discuss this. Additionally, they should discuss the relevance of the third late MVA boost, especially in the light of the strong vector-specific antibody response they showed in an earlier analysis of this study.

Reviewer 2 Report

Authors have designed a study to analyze efficacy of DNA prime/MVA boost vaccine regimens in healthy Tanzanian volunteers for IgG antibodies responses to the HIV-1 Env variable region gp70V1V2. A significantly higher IgG antibodies to V1V2 of subtype AE A244 was observed, followed by antibody responses to subtype C CN54, and low response to subtype B Case A2. Of IgG antibody subclasses, IgG1 responses were found to be dominant over IgG3 and no significant response were observed for IgG2 and IgG4 antibodies.

Though a limited number of vaccinees immunized, the strong point of this study is a long-lasting and durable antibody response conferred against antigen subtype AE (V1V2 CRF01_AE A244 IgG response in 97% four weeks after second MVA boost that was persistent for three years in 75%, further increasing in 95% after third MVA boost). A similar trend was observed for IgG responses to cyclic V2 A244.

Overall, this study is sound and well conceptualized. It would provide a valuable information for development of future vaccine regimens. However, following question and minor points should be addressed.

Is there any data available to distinguish antibody responses in vaccinees receiving HIV-DNA id compared to im?

Minor issues.

Line 127-128, please expand the id to intradermally and im to intramuscularly in Figure 1 legend. It’s already done in text, but it would be nice if it is mentioned in the figure 1 legend as well.

Line 140, should there be a full-stop or a comma after “was used”

In table 2, “positive/total number tested (5)” needs to be changed to “positive/total number tested (%)”

Please mention the positive/total number tested for Fig 2C and 3B

Please provide a reference for line 370-371

Reviewer 3 Report

In this article the authors present specific antibody response evaluation against HIV-1 V1V2 regions in 29 volunteers following immunizations with HIV-1 DNA and HIV-modified vaccinia virus Ankara (MVA) boosts. They show that in a high proportion of vaccinees, vaccination strategy induced durable IgG antibody responses to VIV2 of CRF01_AE A244; a weak anti-V1V2 subtype C responses were also present. They conclude DNA prime/MVA boost HIV vaccine regimen is a vaccination system to beconsider and further evaluated.

This paper is well written and the experimental approach and results are well explained. The weak point of this study is the low number of vaccinees and the durable imnnune response mainly in a CRF virus.

I have no major criticism regarding the experiments presented in this article. Despite statistical analysis, the conclusions are drawn on small numbers. Apart from this, the originality of the results in comparison to their previous article is low (see previous papers and Front. Immunol., 28 April 2020).

Round 2

Reviewer 1 Report

All comments have been sufficiently addressed. Thanks you very much.

Reviewer 3 Report

Though the weak point of the study still remains the limited number of vaccinees immunized, overall, in the present form this paper can be accepted.